# Gene Expression Profiling of Different Huh7 Variants Reveals Novel Hepatitis C Virus Host Factors

**DOI:** 10.3390/v12010036

**Published:** 2019-12-28

**Authors:** Christopher Dächert, Evgeny Gladilin, Marco Binder

**Affiliations:** 1Research Group “Dynamics of Early Viral Infection and the Innate Antiviral Response”, Division Virus-associated Carcinogenesis (F170), German Cancer Research Center (DKFZ), 69120 Heidelberg, Germany; c.daechert@dkfz.de; 2Faculty of Biosciences, Heidelberg University, 69120 Heidelberg, Germany; 3Division Bioinformatics and Omics Data Analytics, German Cancer Research Center (DKFZ), 69120 Heidelberg, Germany; e.gladilin@dkfz-heidelberg.de; 4BioQuant, Heidelberg University, 69120 Heidelberg, Germany

**Keywords:** Hepatitis C virus, HCV, host factor, permissiveness, Huh7, THAP7, NR0B2, CRAMP1, LBHD1, CRYM

## Abstract

Chronic Hepatitis C virus (HCV) infection still constitutes a major global health problem with almost half a million deaths per year. To date, the human hepatoma cell line Huh7 and its derivatives is the only cell line that robustly replicates HCV. However, even different subclones and passages of this single cell line exhibit tremendous differences in HCV replication efficiency. By comparative gene expression profiling using a multi-pronged correlation analysis across eight different Huh7 variants, we identified 34 candidate host factors possibly affecting HCV permissiveness. For seven of the candidates, we could show by knock-down studies their implication in HCV replication. Notably, for at least four of them, we furthermore found that overexpression boosted HCV replication in lowly permissive Huh7 cells, most prominently for the histone-binding transcriptional repressor THAP7 and the nuclear receptor NR0B2. For NR0B2, our results suggest a finely balanced expression optimum reached in highly permissive Huh7 cells, with even higher levels leading to a nearly complete breakdown of HCV replication, likely due to a dysregulation of bile acid and cholesterol metabolism. Our unbiased expression-profiling approach, hence, led to the identification of four host cellular genes that contribute to HCV permissiveness in Huh7 cells. These findings add to an improved understanding of the molecular underpinnings of the strict host cell tropism of HCV.

## 1. Introduction

Hepatitis C virus (HCV) chronically infects more than 70 million people worldwide and is the leading cause of liver cancer in most countries [1,2,3]. For a few years, highly potent direct-acting antivirals (DAAs) have been available, which—under optimal clinical conditions—achieve cure rates above 95% [4]. However, even after successful treatment, patients remain at increased risk to developing hepatocellular carcinoma [5,6,7,8]. In addition, due to the high number of undiagnosed infections and the lack of protection from re-infection, eradication of the disease without further basic research and the development of a prophylactic vaccine is unlikely [9].

HCV belongs to the *Hepacivirus* genus in the *Flaviviridae* family and comprises eight genotypes with at least 86 subtypes [10,11]. The virus particle is enveloped and has a single-stranded positive-sense RNA genome of approximately 9.6 kb, coding for one single open-reading frame that gives rise to 10 mature viral proteins [12]. HCV almost exclusively infects hepatocytes where at least four membrane proteins or receptors are involved in the entry process, i.e., SCARB1 (SR-BI), CD81, CLDN1, and OCLN. Upon receptor-mediated endocytosis, the viral particle is uncoated in a clathrin-dependent manner and the uncapped genome is directly translated by the host translation machinery via its IRES structure [13]. Replication takes place in the cytoplasm at specialized, ER-derived mono-, double-, or multi-membrane vesicles, designated the “membranous web” [14] and is tightly linked to lipid droplets [15]. HCV hijacks many cellular pathways to establish and maintain a productive infection, e.g., autophagy [16] as well as glucose [17,18,19,20] and cholesterol metabolism [21,22]. Recent evidence implies that nuclear receptors contribute to mediating these changes and are, thus, important players during HCV infection [18,23,24].

So far, the only known cell line that robustly supports in vitro replication of HCV is the human hepatoma cell line Huh7 and its derivatives (reviewed in [25]), originally isolated from a 57-year old Japanese male [26]. Strikingly, even within this one cell line, dramatic differences of up to 1000-fold in HCV replication were observed between different passages or subclones, such as Huh7-Lunet [27,28]. It became clear that not only do viral determinants play a role in HCV replication efficiency but, importantly, so do the features of the host cell [27,29,30]. Significant efforts have been taken to understand this strict host cell tropism of HCV, leading to the identification of many important HCV host factors [31,32,33,34,35,36,37], most importantly phosphatidylinositol 4-kinase III alpha (PI4KIIIα; [35,36,37,38,39]), micro-RNA 122 (miR-122; [34,40]), or cyclophilin A [41,42,43,44]. Notably, for technical reasons, most of these factors were identified by knockdown in highly permissive cells and show a reduction of HCV replication in various tested cell lines. In addition to the receptors required for HCV cell entry, only miR-122 is capable of generally increasing HCV replication in certain cell lines, such as HuH6, HepG2, or Hep3B [45]. Another host gene able to increase the replication of HCV is SEC14L2; however, this factor only affects certain HCV strains and, as such, is not sufficient to increase general permissiveness [46]. Thus, although they revealed a great deal of detail about the virus-host interface of HCV, none of these factors can explain the vast differences in HCV replication efficiency between lowly and highly permissive Huh7 cells [27].

We have previously developed a mathematical model able to describe intracellular HCV replication kinetics, both in lowly (Huh7-LP) as well as highly permissive (Huh7-Lunet) cells [47]. By model analysis, we found the assumption of one host cellular factor (HF) involved in the establishment of replication complexes and the membranous web to be sufficient to account for the substantial difference in replication kinetics between these two types of cells. Of note, this HF turned out to be limiting in lowly permissive cells, but abundant and non-limiting in highly permissive cells [47]. This is in full accordance with earlier experimental findings of the Bartenschlager group, who also found that a pro-viral factor limiting replication in lowly permissive cells is much more likely than an inhibitory factor [27]. It should be pointed out that the term “host factor” not necessarily refers to a single protein, but might as well be a more complex host process (e.g., membrane biosynthesis) or refer to a non-proteinaceous component whose abundance may depend on the expression of various host cellular genes. In order to identify this proposed HF or genes contributing to its abundance, we used an unbiased approach and correlated gene expression in eight different uninfected Huh7 cell lines, ranging from lowly to highly permissive, with the respective HCV replication efficiency. By combining different correlation analyses, we identified 34 genes and tested their impact on HCV replication by siRNA-mediated knockdown. We found seven host factors that showed a robust decrease in HCV reporter virus replication as well as in a replicon cell line upon knockdown, i.e., ZNF512B, SFI1, LBHD1 (C11orf48), CRYM, CRAMP1, THAP7, and NR0B2 (SHP). We could further strengthen the implication of some of those genes in HCV replication by overexpressing them in a lowly permissive cell line. Finally, we demonstrated that at least four of them increased HCV replication in this setting and thus fulfilled our requirements of *bona fide* host factors.

## 2. Materials and Methods 

### 2.1. Cell Culture and Cell Lines

All cell lines used in this study are derived from the human hepatoma cell line Huh7 [26]. They were cultured in Dulbecco’s modified Eagle medium (DMEM, Life Technologies, Germany) supplemented with 10% fetal bovine serum (Capricorn Scientific, Ebsdorfergrund, Germany), non-essential amino acids (Life Technologies, Darmstadt, Germany), and 100 μg/mL penicillin and streptomycin. Cells were maintained in a humidified incubator at 37 °C and supplied with 5% CO_2_. Huh7-LucUbiNeo cells used in this study are Huh7 cells stably carrying a subgenomic HCV replicon of genotype 2a (isolate JFH-1), encoding a *Firefly* luciferase reporter and neomycin resistance gene [48]. Naïve, low passage Huh7 cells (passage 10–30) used in this study are referred to as Huh7-LP [29], high passage Huh7 cells (passage 100–150) are referred to as Huh7-HP. Huh7-Lunet and Huh-7/5-2 cells are highly permissive Huh7 cell clones [28]. Huh7-Lunet NP refers to a derivative of Huh7-Lunet, which spontaneously became significantly less permissive during passaging due to unknown reasons [47]. Huh7-Lunet cells used in this study were stably overexpressing the HCV co-receptor CD81 [49].

### 2.2. Gene Expression Profiling and Statistical Analysis

Gene expression data has been generated previously [47]. Briefly, total cellular RNA was extracted from naïve, uninfected Huh7 variants using Trizol according to the manufacturer’s instructions and gene expression was measured using the Affymetrix Human Genome U133 Plus 2.0 platform [47]. Experimental details and microarray data have been deposited at GEO under accession number GSE140114. The core idea of our approach to identify host factors determining HCV permissiveness was to find genes whose expression profile correlates with HCV replication in eight different Huh7 cell lines (Figure 1A). Appendix A shows an example of two genes with either high or low correlation between their expression level and HCV permissiveness in lowly (Huh7 p13, Huh7 p28) and highly (Huh7 Lunet NP, Huh7 p163, Huh7 p154, Huh7/5-2 p44, Huh7 Lunet I, Huh7 Lunet II) HCV replicating cell lines. A straightforward analysis of gene expression patterns is hampered by the fact that gene expression measurements exhibit different levels of statistical significance (detection call A[absent]/P[resent]/M[arginal]). Consequently, not all eight gene expression measurements could always be considered, which lets gene expression cell line patterns become incomplete (see Appendix A). Since one particular numerical criterion may reflect some but disregard other aspects of (dis)similarity between gene expression and HCV replication patterns, five different measures were applied for quantification of the gene impact on HCV replication rate. First, we focused only on genes whose minimum expression level in high HCV replicating cell lines is strictly higher than maximum gene expression in low replicating cells. For characterization of a gene’s relevance to HCV replication, five numerical criterions based on significant Pearson and Spearman correlation between patterns of gene expression and HCV replication in eight Huh7 cell lines as well as particularly high gene expression level in high HCV replicating cells were used (see Appendix A). Specifically, correlation coefficients were calculated between gene expression and log(N), where N denotes the HCV replication efficiency (luciferase activity at 48 h normalized to the input at 4 h) in a particular cell line. Significance of the correlation was assessed according to the threshold *p* ≤ 0.05 for the specific number of statistically significant (i.e., P/M) gene expression measurements in eight different cell lines.

### 2.3. Chemicals and Antibodies

Chemicals and stock solutions used in this study were purchased from Carl Roth (Karlsruhe, Germany) or Sigma-Aldrich (Taufkirchen, Germany) if not stated otherwise. The FXR modulators GW4064 (Hölzel Diagnostika) and (Z)-Guggulsterone (GS) were dissolved in DMSO to 100 and 10 mM, respectively.

### 2.4. Virus Stock Preparation and Titration of JcR2a and Jc1

Infectious full-length viral particles (JcR2a [38], Jc1 [50]) were prepared as described in [21]. In brief, in vitro-transcribed RNA was electroporated into Huh7.5 cells [51] and cell culture supernatants were harvested and sterile-filtered after 24, 48, and 72 h. Jc1 supernatants were further concentrated by incubation with 8% PEG-8000 overnight at 4 °C and centrifugation at 4000× *g*, 30 min, 4 °C. Supernatants were discarded and the pellet was resuspended in complete medium, aliquoted and stored at −80 °C. Virus titers were determined on Huh7.5 cells with an end-point dilution assay (TCID_50_ [52]) using the mouse monoclonal α-NS5A antibody 9E10 (a kind gift from Dr. Charles Rice, The Rockefeller University, New York, NY, USA) in a 1:15,000 dilution. An Excel spreadsheet for calculating the infectious titer is available online [53].

### 2.5. Dengue and Rift Valley Fever Virus Infection

*Renilla* luciferase Dengue reporter virus particles were produced as described previously [54] and Huh7 cells were infected with an MOI of 1 by adding virus particles-containing supernatant. After 48 h, cells were lysed in 100 µL 1× passive lysis buffer (Promega, Mannheim, Germany) and luciferase activity was measured as described below. Rift Valley Fever ∆NSs *Renilla* luciferase reporter virus was described previously [55]. Huh7 cells were infected with an MOI of 0.01 in Opti-MEM^®^ (Promega) for 1 h shaking in a humidified incubator at 37 °C and 5% CO_2_. Medium was then changed to DMEM complete with only 2% fetal bovine serum. Cells were lysed at 48 h in 100 µL 1× Promega Lysis Buffer and luciferase activity was measured as described below.

### 2.6. Reverse Transfection and Infection of Huh7 Cells

Huh7-Lunet-CD81H cells [56] were reverse transfected with 1 pmol siRNA according to the manufacturer’s protocol with slight modifications. In brief, 7.5 × 10^3^ cells were added on top of a transfection mix containing 25 µL Opti-MEM^®^, 0.25 µL RNAiMAX^®^ and 1 pmol siRNA after 10 min incubation in a 96-well format. 24 h later, cells were infected with *Renilla* luciferase reporter virus JcR2a [38] at an MOI of ~0.1. After 96 h, cells were lysed with 1× passive lysis buffer (Promega, Mannheim, Germany) and luciferase activity was measured using a Mithras² LB 943 Luminometer (Berthold Technologies, Bad Wildbad, Germany) with a 480 nm high-sense filter.

### 2.7. Reverse Transfection of Huh7-LucUbiNeo Cells

Cells were reverse transfected with 1 pmol siRNA according to the manufacturer’s protocol with slight modifications. In brief, 1 × 10^4^ cells were added on top of a transfection mix containing 25 µL Opti-MEM^®^, 0.25 µL RNAiMAX^®^ and 1 pmol siRNA after 10 min incubation in a 96-well format. After 96 h, cells were lysed with 1× passive lysis buffer (Promega) and luciferase activity was measured using a Mithras² LB 943 Luminometer (Berthold Technologies).

### 2.8. CellTiter-Glo^®^ Cell Viability Assay

SiRNA cytotoxicity was assessed using the *CellTiter-Glo^®^ Luminescent Cell Viability Assay* kit (Promega). Cells were reverse transfected as described above and mock-infected the next day. After 72 (Huh7-Lunet) or 96 h (Huh7-LucUbiNeo), cells were lysed in a 1:1 mixture of medium and CellTiter-Glo^®^ reagent, incubated for 10 min, shaking at room temperature, and luciferase activity was measured for 1 s using a Mithras² LB 943 luminometer (Berthold Technologies) in white 96-well cell culture plates.

### 2.9. In Vitro-Transcription and Electroporation of RNA into Huh7 Cells

Preparation of sub-genomic HCV reporter RNAs and electro-transfection into Huh7 cells was done as described [57]. Sub-genomic reporter replicons based on HCV genotypes 1b (Con1-ET) and 2a (JFH-1) have been described before (Con1-ET [27], JFH-1 [58]). *SpeI* was used to linearize the Con1-ET plasmid prior to in vitro transcription.

### 2.10. Luciferase Assay

Cells were lysed in 25 µL 1× passive lysis buffer (Promega) on 96-well plates for assessing *Renilla* luciferase activity or 100 µL Luc lysis buffer (1% Triton-X 100, 25 mM glycylglycine, 15 mM MgSO_4_, 4 mM EGTA, 10% glycerol; 1mM DTT added freshly) on 24-well plates for assessing *Firefly* luciferase activity, and frozen at −80 °C. Luciferase activity was measured after thawing lysates in Luc assay buffer (25 mM glycylglycine, 15 mM KPO_4_ buffer, 15 mM MgSO_4_, 4 mM EGTA) containing freshly added 3.36 µM coelenterazine or 10 µM D-luciferin, 2 mM ATP, and 1 mM DTT for *Renilla* or *Firefly* measurements, respectively, using a Mithras² LB 943 luminometer (Berthold Technologies) with a 480 nm high-sense (*Renilla*) or no filter (*Firefly*).

### 2.11. RNA Isolation, Reverse Transcription and qRT-PCR

RNA isolation from cell lysates, reverse transcription and quantitative real-time PCR were performed as described in [59]. Briefly, 1 µg of isolated whole cell RNA was reverse-transcribed with random hexamer primers and gene expression levels were assessed with gene-specific primers and SYBR^®^ Green (Bio-Rad, Feldkirchen, Germany). Primers were designed using Primer3 [60,61,62] or Primer-BLAST [63] and were exon-spanning to avoid amplification of genomic DNA (Table 1). Expression levels were determined using the ∆∆Ct method [64].

### 2.12. Immunoblotting

Roughly 5 × 10^5^ to 1 × 10^6^ cells were collected by centrifugation (350× *g*, 5 min, RT), washed once in 1× PBS and lysed in 60–120 µL 1× sample buffer (16.7 mM TRIS pH 6.8, 5% glycerol, 0.5% SDS, 1.25% β-mercaptoethanol, 0.01% bromophenol blue). A total of 10–20 µL were separated by SDS-PAGE on 8% or 10% polyacrylamide gels and blotted onto PVDF membranes. THAP7 was detected using the THAP7 MaxPab mouse polyclonal antibody (B01) (H00080764-B01, Abnova, Taipei City, Taiwan) in a 1:1000 dilution. NR0B2 was detected using the mouse monoclonal antibody SHP (H-5) (sc-271511, Santa Cruz) in a 1:1000 dilution. Loading controls β-actin and calnexin were detected using a mouse monoclonal β-actin antibody (A5441, Sigma-Aldrich) in a 1:5000 dilution and a rabbit polyclonal antibody (ADI-SPA-865-F, Enzo Life Science, Lörrach, Germany) in a 1:2000 dilution, respectively. Primary antibodies were detected by secondary, horseradish peroxidase-conjugated anti-mouse (1:10,000) or anti-rabbit (1:20,000) antibodies (Sigma Aldrich). Immunoblots were incubated with Clarity ECL substrate (Bio-Rad) and signals were detected with a high-sensitivity CCD camera (ChemoCam Imager 3.2, INTAS, Göttingen, Germany).

### 2.13. Immunofluorescence (IF), Filipin Staining and Bright Field Microscopy

Immunofluorescence was performed as described in [65]. For cholesterol staining, Filipin-III (250 µg/mL, Sigma-Aldrich) was used instead of a primary and secondary antibody. Primary antibodies raised against THAP7 (H00080764-B01, Abnova) or NR0B2 (sc-271511, Santa Cruz) were used in a 1:50 or 1:500 dilution. Goat α-mouse IgG Alexa-Fluor 647 was used as a secondary antibody. For bright field images, cells were seeded and fixed with 4% para-formaldehyde on glass cover slips and mounted using Fluoromount G (Thermo Fisher Scientific, Waltham, MA, USA). Images were captured on a Nikon Eclipse Ti-E microscope (Nikon, Tokyo, Japan) and processed with ImageJ (National Institutes of Health, Bethesda, MD, USA).

### 2.14. Live Cell Imaging Using the IncuCyte System

7500 Huh7-LP, 5000 Huh7-HP or 5000 Huh7-Lunet cells per well were seeded in transparent 96-well cell culture plates in at least quadruplicates and four images per well were taken every 2 h with a 10× objective for at least 72 h using an IncuCyte ZOOM instrument (Essen Bioscience, Ann Arbor, MI, USA). Cell growth was quantified using the “confluency mask” of the IncuCyte ZOOM software (version 2015A or newer). Data is given as the mean and standard deviation from four independent wells.

### 2.15. Generation of Stable Overexpressing Cell Lines

Cell lines stably overexpressing putative HCV host factors were generated as described in [59]. Briefly, lentiviral particles were produced in HEK293T cells [66] by calcium phosphate transfection and used for transduction of target cells. Transduced cells were selected by addition of 5 μg/mL blasticidin or 1 μg/mL puromycin to the culture medium.

### 2.16. Cloning of cDNAs of Putative HCV Host Factors

#### 2.16.1. THAP7, CRYM, and NR0B2

THAP7 (gene ID 80764), CRYM (gene ID 1428; Clone ID 163311861, also known as THBP or DFNA40), and NR0B2 (gene ID 8431, clone ID 183157659; also known as SHP or SHP1) cDNA clones were obtained from the *ORFeome Collaboration (OC) cDNA Clones* [67,68] as pENTR vectors. Note that the THAP7 in this study harbors the natural variation A115P. These constructs were shuttled in pWPI-based expression plasmids under the control of the widely used EF1α [29] promoter for high expression levels or a murine ROSA26 promoter [69] for low expression levels, using the Gateway™ cloning system [70] and sequence-verified prior to use.

#### 2.16.2. LBHD1 (C11orf48)

The 789 bp long LBHD1 (gene ID 79081) cDNA was obtained from gene-specific cDNA prepared from HeLa cell RNA using primer *C11orf48 cDNA 3′-UTR R* (Table 2). This cDNA was used as a template for PCR amplification of the LBHD1 cDNA using primers *C11orf48 attB F* and *C11orf48 attB R open* and the *Q5 HiFi DNA Polymerase* (New England Biolabs) with an annealing temperature of 68 °C. Capital letters indicate attB cloning sites. Note that using these primers results in a cDNA without a stop codon. The resulting PCR product was re-amplified after gel extraction using the *CloneAmp™ HiFi PCR Premix* (Takara Bio). This final PCR amplicon was gel-purified and shuttled into an entry and subsequently into a pWPI-based vector using the Gateway™ cloning system. Sequence integrity was validated by restriction enzyme digest and Sanger sequencing (GATC Biotech, Germany).

#### 2.16.3. CRAMP1

The 3807 bp long CRAMP1 (gene ID 57585; also known as HN1L, TCE4, or CRAMP1L) cDNA was obtained by overlap PCR of two DNA fragments: The first 449 bps were synthesized by BioCat (Heidelberg, Germany) and PCR-amplified using *CRAMP1 attB F* and *CRAMP1 overlap R* (Table 3) with *Q5 HiFi DNA Polymerase* (New England Biolabs) and an annealing temperature of 55 °C. The second fragment was obtained from the *PlasmID* repository at Harvard Medical School (HsCD00342515). The region of interest was PCR amplified from pCR-XL-TOPO vector using primers *CRAMP1 overlap F* and *CRAMP1 attB R open* (Table 3). PCR amplicons were gel-purified using the *NucleoSpin^®^ Gel and PCR Clean-up* (Macherey-Nagel). Both DNA fragments were fused in an overlap PCR using the *CloneAmp™ HiFi PCR Premix* (Takara Bio) with primers *CRAMP1 attB F* and *CRAMP1 attB R open* with an annealing temperature of 55 °C. This final PCR amplicon was gel-purified and shuttled into an entry and subsequently a pWPI-based expression vector as described above.

## 3. Results

### 3.1. Gene Expression Profiling Suggests Putative Novel HCV Host Factors

The replication efficiency of subgenomic HCV reporter replicons differs up to 1000-fold across different clones and passages of the Huh7 cell line (Figure 1A, [27,47]). In a mathematical model of HCV RNA replication we could show that these vastly different replication dynamics can be theoretically explained by the assumption of one single limiting host resource involved in the formation of HCV replication compartments (i.e., the membranous web) that is of limiting abundance in lowly permissive cells, such as low passage Huh7-LP (e.g., p13 or p28 in Figure 1A) [47]. We have previously assessed the replicative capacity, i.e., the “permissiveness“, of eight different passages or subclones of Huh7 cells using a subgenomic luciferase reporter replicon of HCV genotype (gt) 1b (Con1-ET) (Figure 1A). Of note, while the different passages and subclones exhibit somewhat different morphologies (mostly cell size, Appendix A), differences in cell growth were minor (Figure 1B). A basic characterization of different Huh7 passages including the relative capacity of their translation machinery has previously been published and found no obvious relation to HCV replication efficiency [27]. As all eight variants were Huh7-based, we hypothesized that also their overall transcriptome should be highly comparable. In fact, we found this to be the case, with a Pearson correlation coefficient of 0.97 (±0.015) across the whole transcriptomes of all eight cell lines. In order to identify putative HCV permissiveness determining factors, we have previously correlated gene expression levels with HCV replication efficiency using a simple linear model [47]. We have now extended this analysis and used five different approaches of correlation analysis to filter for top scoring hits (see Section 2.2., Appendix A). We found 34 candidates with positive correlation of gene expression and HCV replication efficiency (Figure 1C,D). To determine the role of these genes during HCV replication, we depleted their expression in highly permissive cells (Huh7-Lunet) by siRNA and determined the impact on HCV replication using a reporter virus (JcR2a, Figure 2A).

We tested three siRNAs per candidate gene, and included siRNAs against PI4KIIIα [38] and GFP as positive and negative controls, respectively. We found 14 siRNAs targeting 12 of the 34 tested genes that markedly inhibited JcR2a replication (Figure 2A). To determine whether the depletion of target genes affected cell viability, cytotoxicity of all siRNAs was assessed (Appendix A). Overall, cytotoxic effects were mild and toxic siRNAs reducing cell viability to less than 75% were excluded from subsequent experiments (with the exception of ZNF512B_3). To further validate the contribution of these 12 genes to cellular permissiveness to HCV, we depleted their expression in Huh7-LucUbiNeo cells, stably replicating a subgenomic HCV (gt2a) replicon [48,71,72]. While we observed the same tendency for most of the knocked-down genes, effect sizes were generally smaller as compared to the JcR2a infection assay. This is in line with the mathematical model that suggests involvement of the limiting host factor in the establishing of the replication organelle; in the stable replicon cell line, HCV has already profoundly established its replication compartment, while it needs to be formed de novo in the infection assay. Furthermore, the stable Huh7-LucUbiNeo cell line and the contained replicon genome have gone through stringent selection and adaptation in the course of multiple rounds of passaging. Still, we could confirm seven of the 12 targeted genes to significantly affect HCV replication also in this setting (Figure 2B). We selected the five genes with the greatest impact on HCV replication for further validation: LBHD1 (also called C11orf48), THAP7, CRAMP1, NR0B2 (also called SHP), and CRYM (colored bars in Figure 2B).

### 3.2. Overexpression of Host Factors in Lowly Permissive Cells Increases HCV Replication

Our mathematical model, in agreement with previous biological experiments [27], suggested that host factor abundance is limiting in lowly permissive cells [47]. In order to test if our identified candidate genes fulfill this requirement of a *bona fide* host factors, we tested whether their overexpression in lowly permissive cells was able to rescue HCV replication efficiency. For that purpose, we cloned the cDNAs of the five top genes into a lentiviral vector and stably transduced lowly permissive Huh7-LP cells. All five genes were robustly overexpressed in mRNA but also (exception: NR0B2-FLAG) at the protein level (Appendix A). We then electro-transfected the stable cell lines with in vitro-transcribed subgenomic HCV luciferase reporter replicon RNA of the cell culture-adapted gt1b strain Con1-ET [71,72], or infected cells with the JcR2a gt2a full-length reporter virus and recorded replication over time.

### 3.3. CRYM, LBHD1, and THAP7 Increase HCV Replication in Lowly Permissive Huh7 Cells

Despite promising effects in the knockdown screen, overexpression of CRAMP1 did not significantly affect replication of the subgenomic replicon, neither with an N-terminal HA-, nor with a C-terminal FLAG-tag (Figure 3A,B). However, it raised JcR2a virus replication measurably above the background, which is hardly detectable in empty vector control cells (Figure 3C). It should be noted that CRAMP1 overexpression levels were the lowest observed for all candidate genes (Appendix A). Overexpression of HA-tagged CRYM [73] in Huh7-LP cells resulted in a slight but reproducible increase in replication of the subgenomic HCV replicon after 72 h (Figure 3D), as well as a notable replication upon infection with the JcR2a reporter virus (Figure 3F). However, when tagged at the C-terminus with a FLAG-tag, HCV replication was not affected (Figure 3E), despite substantial overexpression as assessed by qRT-PCR and Western blot (Appendix A). Similarly, overexpression of HA-LBHD1, a barely characterized gene, resulted in a statistically significant increase in subgenomic HCV replication (Figure 3G) as well as JcR2a infection (Figure 3I). Nonetheless, also for LBHD1, C-terminal tagging with FLAG abolished this positive effect on HCV and even led to a slight decrease in HCV replication (Figure 3H). It was surprising to see these significant differences between the differently tagged variants of CRYM and LBHD1, which were reproducibly observed across three independent repetitions of the experiments. We cannot rule out that C-terminal tagging with the FLAG-epitope compromises these proteins’ biological function or interferes with critical (viral or cellular) interaction partners.

The strongest and most robust effect on HCV replication efficiency was observed for overexpression of THAP7. Huh7-LP cells were stably transduced with HA-THAP7 and successful expression was confirmed by qRT-PCR (Appendix A), immunoblotting (Appendix A) and immunofluorescence staining (Appendix A). Overexpression of THAP7 resulted in a highly significant 5.5-fold increase (at 72 h) in replication of the subgenomic HCV replicon (Figure 4A), and successfully rescued HCV replication by one log upon infection with JcR2a (Figure 4B). This was the strongest enhancement of host cell permissiveness we have observed so far, and it was reproducible also with the C-terminally tagged THAP7-FLAG construct (Appendix A). As our prediction for a limiting host factor is that it is abundant and non-limiting in highly permissive Huh7-Lunet cells, we furthermore tested the effect of THAP7 expression in this cell line. Despite robust overexpression (Appendix A), HCV replication was not at all affected by THAP7 in Huh7-Lunet cells (Figure 4C). Therefore, while neither CRAMP1, CRYM, LBHD1, nor THAP7 alone can explain the huge difference in HCV replication between Huh7-LP and -Lunet cells, these four genes, and in particular THAP7, are some of the very few genes described so far, whose expression could be shown to directly contribute to HCV permissiveness of Huh7 cells.

### 3.4. NR0B2 Expression Levels Sensitively Modulate HCV Replication Efficiency

We confirmed the overexpression of HA-NR0B2 in Huh7-LP cells by qRT-PCR (Appendix A), immunoblotting (Appendix A) and immunofluorescence staining (Appendix A). NR0B2 showed a mainly nuclear localization consistent with its described role as a nuclear receptor and transcriptional repressor, but also a diffuse signal with some rare speckles in the cytoplasm (white arrows, Appendix A). This partially cytoplasmic localization is in line with the reported finding of direct interaction between NR0B2 and HCV NS5A, an important ER-membrane-associated component of the viral replication machinery [74]. Unexpectedly, overexpression of HA-NR0B2 in lowly permissive Huh7-LP cells lead to a significant 3.4-fold decrease in HCV replication (Figure 5A); the same was observed for a C-terminally FLAG-tagged version of NR0B2 (Appendix A). This was surprising as both our initial correlation analysis (Figure 1) as well as the siRNA knockdown in Huh7-Lunet cells (Figure 2) suggested that NR0B2 levels positively correlate with HCV replication efficiency. One possible explanation could be that the expression levels reached in our overexpression system were substantially higher than the levels found in highly permissive cells, possibly posing a secondary disadvantage on HCV. In fact, Huh7-Lunet cells expressed only 2.8-fold more NR0B2 than Huh7-LP cells (Figure 6B, black/white bar), whereas lentiviral overexpression yielded 850-fold higher mRNA levels (Appendix A). To test this hypothesis, we overexpressed NR0B2 in Huh7-Lunet cells, which resulted in similar expression levels and localization (Appendix A) as for overexpression in Huh7-LP cells. Strikingly, NR0B2 overexpression in these *a priori* highly permissive cells resulted in a dramatic 99.5% reduction in HCV RNA levels (Figure 5C). To see if this effect is genotype-specific, we tested a subgenomic replicon based on the gt2a isolate JFH-1 [75] alongside our gt1b Con1-ET replicon. The deleterious effect of HA-NR0B2 overexpression was also observed for the gt2a replicon, however to a less pronounced extent (Figure 6A), likely due to substantially higher replication levels and generally less sensitivity towards host cellular determinants of this isolate. These findings supported the hypothesis that exceeding NR0B2 expression beyond a certain optimum may lead to adverse effects. To corroborate this notion, we diluted the lentiviral particles prior to transduction of Huh7-Lunet cells to reach lower expression levels. This approach led to significantly less NR0B2 mRNA expression (180- and 63-fold over endogenous levels, Figure 6B) and, in fact, these lower expression levels resulted in increased levels of HCV RNA replication as compared to our initial overexpression (Figure 6C). However, even in the lower expressing sample, NR0B2 still negatively impacted on HCV replication. Thus, we made use of an artificial promoter based on the murine ROSA26 locus, which leads to very weak but stable expression in human cells. Using this promoter, we reached expression levels of only eight-fold above the control (Figure 6B). This very mild increase in protein expression indeed increased replication of the gt1b replicon slightly but statistically significantly in Huh7-Lunet cells (Figure 6C). These findings confirmed our hypothesis that HCV is very sensitively dependent on optimal NR0B2 expression, where both too little as well as too much protein is detrimental for viral replication (model illustration in Figure 6D). We could confirm this strict dose-dependence in HCV infection using the JcR2a reporter virus, whereas replication of the related Dengue virus (DENV) was only marginally affected and the unrelated Rift valley fever virus (RVFV) was unaffected altogether (Figure 7). Interestingly, although to a much lesser extent, DENV replication showed a similar optimum behavior for NR0B2 levels as observed for HCV. Although high levels of NR0B2 have mild effects on cell growth and alter cell morphology slightly (Appendix A), the fact that DENV and RVFV are hardly affected by NR0B2 overexpression support the notion of an HCV-specific effect.

### 3.5. The FXR-NR0B2 Axis Regulates HCV Replication through Bile Acid and Cholesterol Homeostasis

NR0B2 (also known as SHP) is a well-established downstream effector of farnesoid X receptor FXR (NR1H4) signaling (Figure 8A), and it has previously been shown that bile acids can increase replication of HCV in an FXR-dependent manner [76,77,78]. Additionally, in our hands, activation of FXR with its agonist GW4064 and inhibition by the antagonist (Z)-Guggulsterone led to an increase and decrease in HCV replication, respectively (Figure 8B). We hypothesized that activation of FXR might emulate our slight overexpression of ROSA26:NR0B2 in Huh7-Lunet cells. It appears feasible that the negative effect of NR0B2 overexpression is mediated via regulation of FXR. Although there is no reported evidence for a direct transcriptional regulation of FXR by NR0B2, we observed a negative correlation between FXR and NR0B2 mRNA levels in our overexpressing cell lines (Pearson coefficient −0.79; Figure 8C). While this might indeed explain the inhibitory effect on HCV replication, it might rather be the indirect consequence of blocked bile acid synthesis by high levels of NR0B2 (Figure 8A, [79,80]). In support of this notion, we found less than 50% of total bile acids in cell lysates from Huh7-Lunet cells overexpressing HA-NR0B2 compared to the HA-eGFP control (Appendix A); however, absolute levels of bile acids were at the lower limit of detection even for the control cells. On the other hand, NR0B2 regulates bile acid anabolism by transcriptional inhibition of the cholesterol-hydroxylase CYP7A1 and cholesterol is known to play a crucial role in the HCV life cycle as well [21]. Thus, we also examined the distribution of free cholesterol in the cell using Filipin-III [81]. In contrast to a very defined, speckled ER-like localization of cholesterol in control cells, NR0B2 overexpression led to substantial relocalization of cholesterol away from the perinuclear region towards the plasma membrane of the cell (Figure 8D). These findings suggest that a finely balanced expression of NR0B2 might be required for maintaining sufficient levels of cholesterol at the sites of replication of HCV.

## 4. Discussion

HCV exhibits a very narrow host tropism, with only humans and chimpanzees being susceptible to infection. However, even within the human system, the virus efficiently infects only hepatocytes, with reports on extrahepatic replication being scarce [82]. Astonishingly, even in human hepatocytes, ex vivo replication of HCV is exceedingly difficult—isolated primary hepatocytes do support infection and replication of highly efficient HCV isolates, such as JFH-1, however, replication is weak and transient [83]. While this may be due to an intact antiviral response in primary cells, HCV fails to replicate in the vast majority of cultured hepatocytic cell lines, despite their antiviral system oftentimes being highly attenuated. In fact, only the Huh7 cell line—and especially some derived clones and subclones—proved to be robustly permissive to HCV in vitro [25]. This extreme degree of host cell selectivity is in contrast to even rather closely related viruses, such as the Flaviviruses [84,85,86], and likely reflects HCV’s finely tuned virus-host-interface that also permits the virus to replicate persistently within a cell.

In order to better understand this well-balanced replication of HCV, we have previously developed a mathematical model describing intracellular RNA replication dynamics of HCV in Huh7 cells of high and low replicative capacity [47]. As both, highly and lowly permissive cells were based on Huh7 and, hence, genetically highly homologous, we speculated that only a single host-dependent step along the replication cycle might be affected, limiting replication efficiency in lowly permissive cells. By simulating host-dependence at every conceivable step along the replication cycle, we could identify one model in which the *a priori* concentration of one single host factor species sufficed to explain the substantially different replication dynamics between highly and lowly permissive cells. Interestingly, this host factor was limiting the replicative capacity only in lowly permissive cells, whereas it was predicted to be non-limiting in Huh7-Lunet cells [47]. This prediction is in line with a previous study, which found a pro-viral resource of limited abundance to be more likely to govern Huh7 permissiveness than an inhibitory factor [27]. Nonetheless, we cannot fully rule out a negative regulator to actively inhibit HCV replication in lowly permissive cells, such as an enhanced stress response or a more potent antiviral response. While the latter would be an intriguing possibility [87], we have extensively investigated this previously and could not find a significant contribution of IRF3 signaling to the HCV permissiveness phenotype of Huh7 [29]. We have, therefore, in this study focused on a positive host factor present in highly permissive cells. However, although mathematically it was sufficient to assume one single “species” in the replication model to be limiting [47], this does not necessarily imply a single gene or gene product is responsible for the full effect. Instead, “host factor” might comprise a complex cellular process giving rise to a limiting resource, which could be membranes (for building the membranous web) or specifically localized cholesterol, PI4P or other lipidic components essential for proper functioning of the HCV replicative machinery [88]. In this scenario, numerous genes might contribute to the permissiveness phenotype.

Most previously published efforts to identify host cellular determinants that HCV depends on, so called “host dependency factors”, are based on the depletion of cellular genes by siRNA [89,90] or, very recently, by CRIPSR/Cas9-mediated genome editing [91,92]. Such approaches are unsuited to identify factors naturally limiting HCV replication in non- or lowly permissive cells. The group of Charles Rice, therefore, took an inverse approach, overexpressing a comprehensive human cDNA library in Huh7.5 cells and screening for increased replication efficiency [46]. That study identified one single gene, SEC14L2, overexpression of which increased replication of otherwise weakly replicating HCV isolates. However, the authors found this factor to increase replication of these isolates across lowly and highly permissive Huh7 variants as well as another liver cell line, Hep3B/miR122; SEC14L2, therefore, does not seem to be a determinant of the strongly varying permissiveness of different Huh7 lines.

In our present study, we attempted to identify host cellular genes contributing to the permissiveness phenotype by directly comparing full-genomic expression profiles across a panel of eight differently permissive Huh7-based cell lines, as we had already suggested in our previous publication [47]. For this purpose, we correlated gene expression levels with HCV replication efficiency in the respective cell lines using the combination of five different criteria. We found 34 genes to show a robust positive correlation and, hence, went on to experimentally challenge their role as a permissiveness determining factor. Confirming the utility of our expression profiling approach, roughly one third of the candidate genes turned out to in fact have an impact on HCV replication when knocked down. Using a second, independent and somewhat more stringent experimental system, we confirmed five genes to robustly impede HCV replication when knocked down: CRAMP1, CRYM, LBHD1, NR0B2, and THAP7.

CRAMP1 is barely characterized, but recent studies found its homolog in *Drosophila* to modulate gene expression by altering epigenetic marks [93]. It is likely that the negative effect its knockdown had on HCV replication is the result of a dysregulation of one or several of its target genes. However, overexpression in lowly permissive Huh7 cells had no effect on replication of a subgenomic HCV replicon, and only slightly increased replication of the reporter virus JcR2a. We, hence, have not studied this gene further. LBHD1 (C11orf48) is another poorly studied gene of unknown function. Overexpression in Huh7-LP cells increased replication of a subgenomic gt1b reporter replicon by more than two-fold and also markedly boosted replication upon infection with a gt2a reporter virus (JcR2a). Hence, LBDH1 is one of very few genes whose expression can increase HCV replication in Huh7-LP cells and it likely contributes to cellular permissiveness for HCV. CRYM yielded the strongest impact on HCV replication in the Huh7-LucUbiNeo replicon cell line when knocked down. Upon overexpression in Huh7-LP cells, it had a moderately positive effect on subgenomic gt1b RNA replication, and strongly increased full-length gt2a reporter virus replication. Functionally, CRYM was very recently reported in mice on high-fat-diet to be linked to modulation of nuclear receptor PPARγ activity, which has previously been shown to also affect HCV replication [94]. However, reports are divergent as to whether PPARγ acts in a pro- or anti-viral manner in HCV [24]. It should be noted that C-terminally FLAG-tagged variants of both, LBHD1 and CRYM, failed to reproduce the effect on HCV replication; however, as the N-terminally HA-tagged versions showed robust and reproducible effects in two different HCV genotypes, we assume the C-terminal tag might interfere with proper protein function or with important protein-protein interactions.

For THAP7, we found its knockdown to have a highly significant impact on HCV replication, both in the infection assay as well as in stable replicon cells. Notably, overexpression in lowly permissive Huh7-LP cells was able to rescue HCV replication by more than five-fold in the subgenomic replicon system and by more than 10-fold in JcR2a infection, whereas overexpression in highly permissive Huh7-Lunet cells had no effect at all, compatible with a role as a limiting host factor in the sense of our model prediction. While the observed effect falls short of explaining the full difference in HCV replication between Huh7-LP and Huh7-Lunet cells (10- to 1000-fold), THAP7 constitutes—to the best of our knowledge—the one single gene that most substantially determines cellular permissiveness to HCV replication across different Huh7 variants. Functionally, this is likely not a direct effect on (or interaction with) the HCV replicative machinery. THAP7 is a chromatin-associated transcriptional repressor that binds to H3 and H4 histone tails, recruits chromatin modifiers, like histone deacetylase 3 (HDAC3) or nuclear receptor co-repressor 1 (NCOR1), and leads to repression of gene expression via deacetylation of target loci [95,96]. It is conceivable that THAP7 recruits HDAC3 to genes restricting HCV replication, as it has been shown that HDAC3 inhibitors suppress HCV replication in Huh7 cells and in a mouse model [97,98]. This in line with our finding in that THAP7 silencing suppresses HCV replication in Huh7-Lunet cells. Future studies should identify genes regulated by THAP7 and assess their individual impact on the HCV life-cycle in order to further our mechanistic understanding of host cellular requirements for HCV replication.

Knockdown of the atypical orphan nuclear receptor NR0B2 (alias SHP), in contrast to one previous report [17], significantly decreased HCV replication in our hands, both, upon infection in Huh7-Lunet cells, as well as in the Huh7-LucUbiNeo stable replicon cell line. Unexpectedly, we also found that overexpression in lowly permissive cells always led to a significant decrease in viral replication, seemingly contradicting the knockdown results. We, therefore, also overexpressed NR0B2 in Huh7-Lunet cells to see if our expression construct caused a general inhibition of viral replication even in cells that generally very robustly replicate HCV to high levels. Strikingly, overexpression of NR0B2 in these highly permissive cells resulted in a dramatic 99% drop in HCV replication. Taken together with the positive correlation of expression levels in lowly to highly permissive cells, and with the results of the knockdown experiments, this strongly negative impact of overexpression suggested an important role of NR0B2 for HCV replication, however, following a non-intuitive dose-response relation. In fact, our results using a panel of increasingly strong overexpression support a model, in which expression in Huh7-LP cells is critically limiting—as suggested by the correlation analysis—and reaches an optimum at the levels found in Huh7-Lunet cells. However, at even higher levels, such as that achieved by ectopic overexpression, NR0B2 unfolds negative effects sensitively impacting HCV replication (optimum-dose model, see Figure 6D). Unfortunately, we were technically not able to achieve slight enough expression levels to observe an increase in replication for lowly permissive cells, which would be the actual evidence for a role of NR0B2 as another permissiveness determining host factor in Huh7 cells. Still, this drastic effect of increasing expression levels beyond the levels found in highly permissive cells is striking and highlights once again the finely tuned virus-host interface of HCV as NR0B2 levels seemed not to significantly impact DENV or RVFV replication. In its sensitive optimum behavior it is reminiscent of the lipid kinase PI4KIIIα, whose activity is limiting and needs to be stimulated by HCV in primary hepatocytes, whereas PI4KIIIα is highly expressed in Huh7 cells and further stimulation leads to unfavorable conditions for HCV replication. The lab of Volker Lohmann has shown that cell culture (i.e., Huh7)-adapted strains of HCV have, thus, evolved to lose PI4KIIIα stimulating activity in order to cope with the situation found in these cells, and non-adapted HCV variants can only replicate in Huh7 cells upon (siRNA-mediated or pharmacological) inhibition of PI4KIIIα [99]. In line with a previous report [74], which further showed a direct interaction of NS5A with NR0B2, we have found our cell culture-derived HCV to mildly induce expression of NR0B2 (see Appendix A); however, conflicting observations have been published as well [17,100]. In the light of these findings, it will be very interesting to investigate the situation in primary human hepatocytes and non-adapted HCV isolates in future studies.

Mechanistically, as for THAP7, it is likely that NR0B2 effects on HCV are indirect. NR0B2 is a nuclear receptor with no known ligand; however, it harbors a ligand-binding domain with two conserved motifs allowing the interaction with and regulation of several other nuclear receptors [101,102]. In the case of PPARγ, NR0B2 is known to bind to its promoter and activate transcription of this nuclear receptor [103,104]. As described above, PPARγ activation has been implicated with HCV replication, but literature reports are not entirely coherent. Recently, other nuclear receptors have also been shown to be dysregulated in HCV infection and partially usurped by the virus, e.g., PPARα, LXR, RXR and FXR [18,24]. Particularly FXR also has a strong link to NR0B2, acting upstream of it and inducing its expression as part of a negative feedback loop in the bile acid metabolism in liver cells; while FXR gets activated by bile acids, NR0B2 inhibits bile acid synthesis (see Figure 8A) [79,80]. In fact, activating FXR by a specific, non-steroidal agonist (GW4064) or inhibiting it by the FXR antagonist Z-Guggulsterone mildly increased or significantly inhibited HCV replication, respectively (Figure 8B) [77], mimicking the effects of modulated NR0B2 expression. Interestingly, we found indications that also the opposite might be true, in that NR0B2 feeds back on FXR expression, which has not been reported before. In our panel of Huh7 cells stably overexpressing NR0B2 to different levels, we found a negative correlation between NR0B2 expression levels and FXR mRNA levels, and, additionally, bile acid levels (Appendix A). This might indicate a relatively complex double feedback type of mutual regulation of FXR and NR0B2, which would also be in line with our observation of a highly sensitive dose-response behavior in overexpressing NR0B2 with regard to HCV replication. This could also explain why treatment of lowly permissive Huh7-LP cells with the FXR agonist GW4064, in contrast to Huh7-Lunet, did not rescue HCV replication (not shown), as these cells not only have a substantially lower expression of NR0B2, but also of FXR (four-fold less, see transcriptome data). We, hence, propose that correlation of NR0B2 expression levels with the permissiveness of cells towards HCV is based on fine-tuning of the whole system of bile acid production and signaling through FXR and NR0B2. Still, the mechanistic underpinnings of how this system affects HCV replication remain elusive. An intriguing, but speculative link to the HCV replicative machinery is cholesterol. On the one hand, it is well established that NR0B2 inhibits bile acid synthesis by downregulating the expression of the crucial cholesterol hydroxylase/monooxygenase CYP7A1 [79,80]. On the other hand, over the last ten years it became increasingly evident that HCV vitally depends on proper distribution of cholesterol in the cell and particularly in the ER-membrane, from which HCV derives its replicative membrane system (“membranous web”) [21,105,106]. In fact, we found NR0B2 overexpression to strongly redistribute cellular cholesterol away from ER-like perinuclear structures towards the plasma membrane. This substantial depletion of cholesterol from endomembranes plausibly suffices to explain the drastic impact NR0B2 overexpression has on HCV replication; however, further mechanistic studies are warranted.

## 5. Conclusions

HCV has a very narrow host cell tropism, not only restricted by entry receptors but also by different cellular functions crucial for the replication of its RNA genome. Even within the most robust cell culture model, the Huh7 cell line, HCV replicates to vastly varying levels in different variants of the Huh7 cell line. Understanding which factors are responsible for low or high replication efficiency might contribute to developing novel cell culture systems or even small animal models and devising future generations of host-directed antiviral regimens. By correlating gene expression profiles of eight different Huh7 variants to their respective capacity to replicate HCV, we successfully identified novel HCV host factors. At least four genes, CRYM, LBHD1, THAP7, and NR0B2, dictate HCV replication efficiency dependent on their expression levels. The most striking effects were observed for THAP7 and NR0B2, both of which are transcriptional regulators and can lead to changes in expression of an unknown number of target genes. NR0B2 appears to be critically involved in the cholesterol and bile acid homeostasis of host cells, and its excessive expression led to a dramatic 2-log reduction of HCV replication. With our unbiased transcriptional profiling approach, we have hence succeeded to identify novel host genes significantly contributing to the differential permissiveness towards HCV replication seen across different closely related cell lines. We could, hence, confirm the prediction of our mathematical HCV replication model that lowly permissive Huh7 are lacking host factors required for efficient replication. Our data may furthermore contribute to a better understanding of the molecular underpinnings of HCV host cell tropism and, thereby, to the development of improved model systems for this disease, which will continue to be important until affordable antivirals or, ideally, a prophylactic vaccine become available.

## Figures and Tables

**Figure 1 viruses-12-00036-f001:**
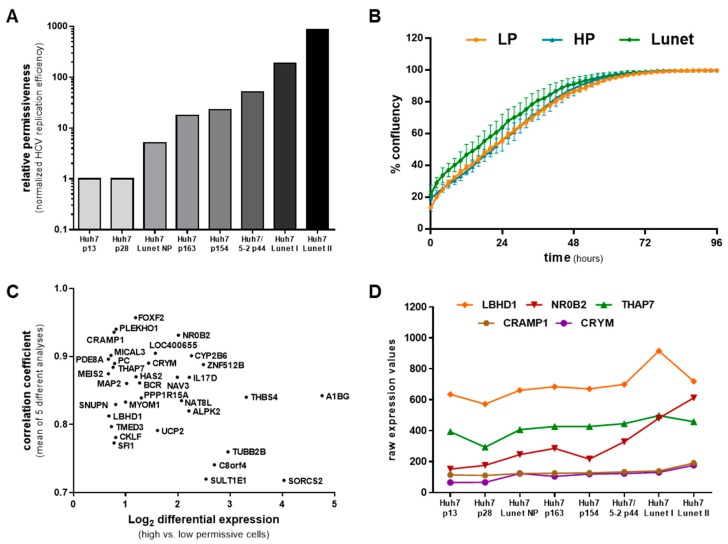
Gene expression profiling and statistical analysis revealed potential HCV permissiveness determining factors. (**A**) HCV (gt1b) replication efficiency differs up to 1000-fold in different passages and clones of the Huh7 cell line. This panel was adapted from [47]); “HCV replication efficiency” refers to the ratio of HCV reporter levels at 48 h to input levels at 4h, see methods. Data was normalized to the two lowly permissive passages Huh7-p13 and -p28. Data represents only one biological replicate measured in parallel to the preparation of total cellular RNA used for microarray experiments; variance across three replicate wells was negligible, hence error bars were omitted. (**B**) Huh7-LP (LP), -HP (HP), or -Lunet (Lunet) cells were seeded in 96-well plates and imaged every 2 h using the IncuCyte system. Data represents mean and standard deviation from four independent wells. (**C**) Correlation analyses of expression profiles and HCV replication efficiency across all eight different naïve Huh7 cell lines revealed 34 potential HCV host factors (see also Appendix A). (**D**) Raw expression data of the followed-up candidates in all eight tested different Huh7 cell lines.

**Figure 2 viruses-12-00036-f002:**
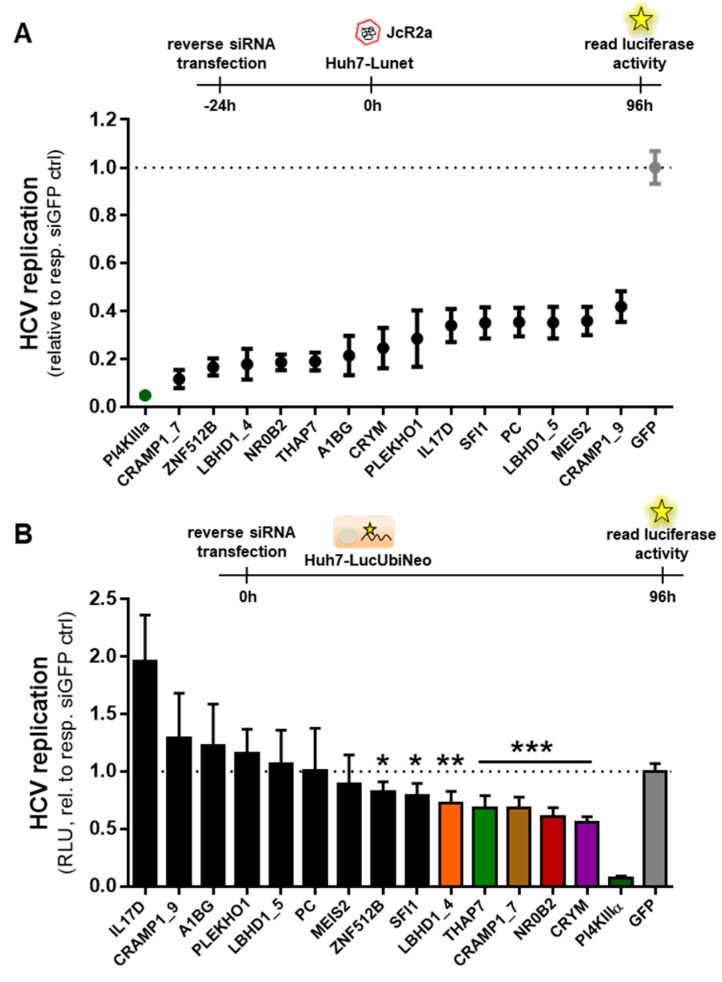
SiRNA-mediated knockdown of potential host factors reduced HCV replication in an infection (**A**) and replicon model (**B**) in highly permissive cells. (**A**) Huh7-Lunet cells were reverse transfected with siRNAs against the indicated genes for 24 h prior to infection with the *Renilla* luciferase reporter virus JcR2a (gt2a) for 72 h. Data is normalized to siRNA targeting GFP and shown are means and standard deviations from three independent biological experiments. (**B**) Huh7-LucUbiNeo cells (gt2a, subgenomic replicon) were reverse transfected with siRNA and luciferase activity was measured 96 h later. Data is normalized to siRNA targeting GFP and shown are means and standard deviations from two independent biological experiments. Statistical significance was calculated using two-tailed, non-paired Student’s *t*-test. * *p* ≤ 0.005, ** *p* ≤ 0.001, *** *p* ≤ 0.0001.

**Figure 3 viruses-12-00036-f003:**
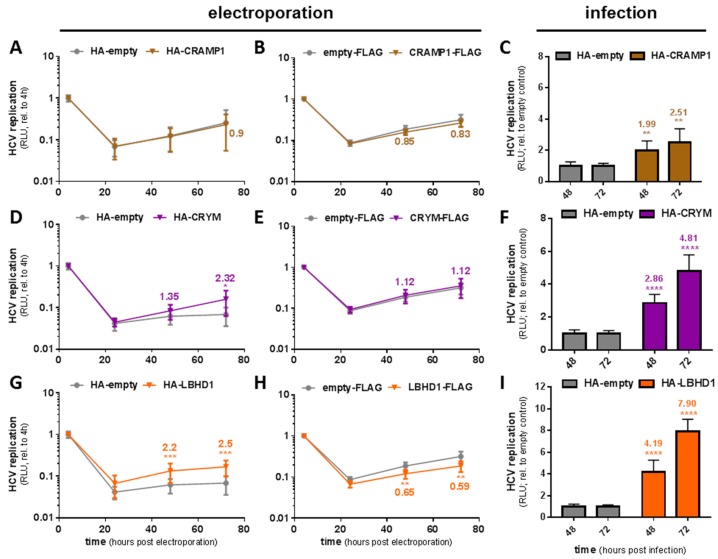
Overexpression of potential permissiveness determining factors in lowly permissive Huh7-LP cells. (**A**,**B**) Replication of a subgenomic HCV luciferase reporter replicon (gt1b) after electro-transfection into HA-CRYM (**A**) or CRYM-FLAG (**B**) overexpressing Huh7-LP cells compared to the respective empty vector control in a time-course of 72 h. (**C**) Expression levels of CRYM mRNA determined by qRT-PCR in CRYM-FLAG overexpressing Huh7-LP cells versus cells transduced with an empty-FLAG vector. (**D**,**E**) Replication of HCV in HA-LBHD1 (**D**) or LBHD1-FLAG (**E**) overexpressing Huh7-LP cells and (**F**) according expression levels of LBHD1. (**G**,**H**) Replication of HCV in HA-CRAMP1 (**G**) or CRAMP1-FLAG (**H**) overexpressing Huh7-LP cells and (**I**) according expression levels of CRAMP1. Luciferase data is normalized to the input signal four hours post electro-transfection and shows mean and standard deviation from at least three independent biological experiments. Significance was calculated using two-tailed, non-paired Student’s *t* test. * *p* ≤ 0.05, ** *p* ≤ 0.005, *** *p* ≤ 0.001, **** *p* ≤ 0.0001.

**Figure 4 viruses-12-00036-f004:**
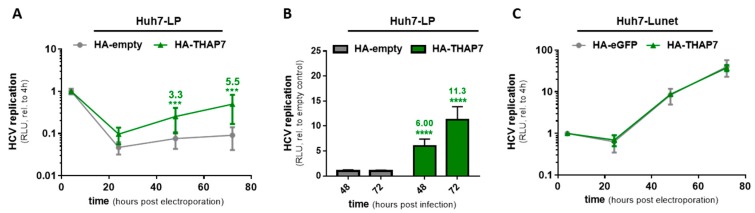
THAP7 is a limiting host factor for HCV replication in lowly but not in highly permissive Huh7 cells. (**A**) Replication of a subgenomic luciferase reporter replicon (gt1b) after electro-transfection into HA-THAP7 overexpressing Huh7-LP cells compared to an HA-empty vector control in a time-course of 72 h. (**B**) Full-length *Renilla* luciferase reporter virus (JcR2a) replication in HA-THAP7 overexpressing Huh7-LP cells compared to an HA-empty vector control after 48 and 72 h. (**C**) Replication of a subgenomic luciferase reporter replicon (gt1b) after electro-transfection into HA-THAP7 overexpressing Huh7-Lunet cells. HCV replication in (**A**,**C**) was normalized to the input signal four hours post transfection and (**A–C**) show mean and standard deviation from at least three independent biological experiments. Significance was calculated using two-tailed, non-paired Student’s *t*-test. *** *p* ≤ 0.001, **** *p* ≤ 0.0001.

**Figure 5 viruses-12-00036-f005:**
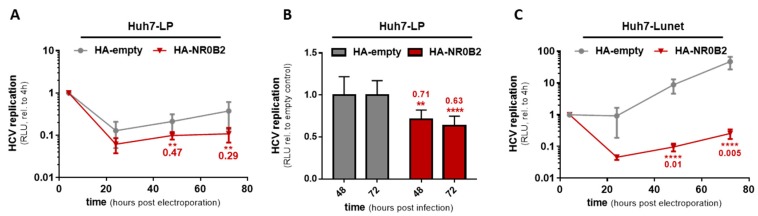
NR0B2 overexpression inhibits HCV replication in lowly and highly permissive Huh7 cells. (**A**) Replication of a subgenomic luciferase reporter replicon (gt1b) after electro-transfection into HA-NR0B2 overexpressing Huh7-LP cells compared to an HA-empty vector control in a time-course of 72 h. (**B**) Full-length *Renilla* luciferase reporter virus (JcR2a) replication in HA-NR0B2 overexpressing Huh7-LP cells compared to an HA-empty vector control after 48 and 72 h. (**C**) Replication of a subgenomic luciferase reporter replicon (gt1b) after electro-transfection into HA-NR0B2 overexpressing Huh7-Lunet cells compared to an HA-eGFP vector control in a time-course of 72 h. Luciferase data (**A**,**C**) is normalized to the signal four hours post electro-transfection and (**A–C**) show mean and standard deviation from at least three independent biological experiments. Significance was calculated using two-tailed, non-paired Student’s *t*-test. ** *p* ≤ 0.005, **** *p* ≤ 0.0001.

**Figure 6 viruses-12-00036-f006:**
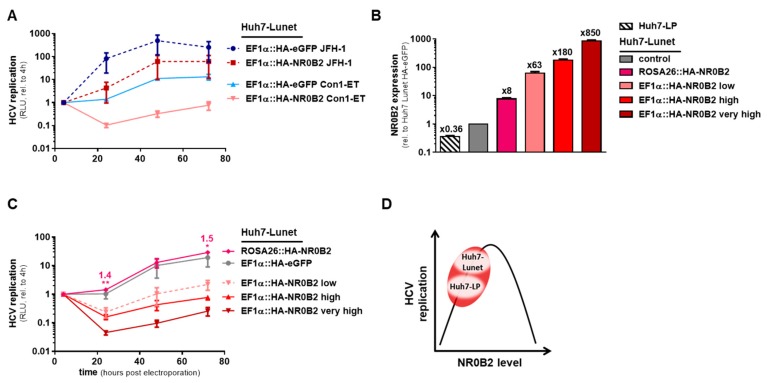
NR0B2 expression levels determine HCV replication efficiency independent of the genotype. (**A**) Replication of subgenomic luciferase reporter replicons of gt2a (JFH-1) or gt1b (Con1-ET) after electro-transfection into Huh7-Lunet overexpressing HA-NR0B2 compared to an HA-eGFP expressing control in a time-course of 72 h. (**B**) Expression levels of Huh7-Lunet cells expressing different amounts of HA-NR0B2 compared to an HA-eGFP expressing control cell line by qRT-PCR. (**C**) Replication of a subgenomic luciferase reporter replicon (gt1b) after electro-transfection into Huh7-Lunet cells expressing different amounts of HA-NR0B2 compared to an HA-eGFP control in a time-course of 72 h. (**D**) Proposed model of the correlation between NR0B2 levels and HCV replication efficiency in Huh7-Lunet and -LP cells. Luciferase data (**A**,**C**) is normalized to the input signal four hours post transfection and shows mean and standard deviation from at least three independent biological experiments. Significance was calculated using two-tailed, non-paired Student’s *t*-test. * *p* ≤ 0.05, ** *p* ≤ 0.005.

**Figure 7 viruses-12-00036-f007:**
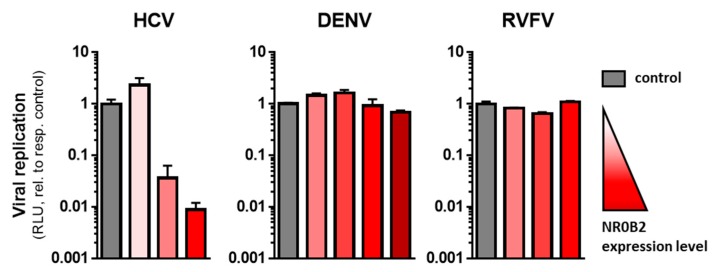
NR0B2 specifically affects HCV replication. Huh7-Lunet cells overexpressing different levels of NR0B2 (see Figure 6B) were infected with HCV, DENV, or RVFV luciferase reporter viruses, respectively. Luciferase activity was measured after 72 h for HCV and 48 h for DENV and RVFV, respectively. Note the logarithmic *y*-axis.

**Figure 8 viruses-12-00036-f008:**
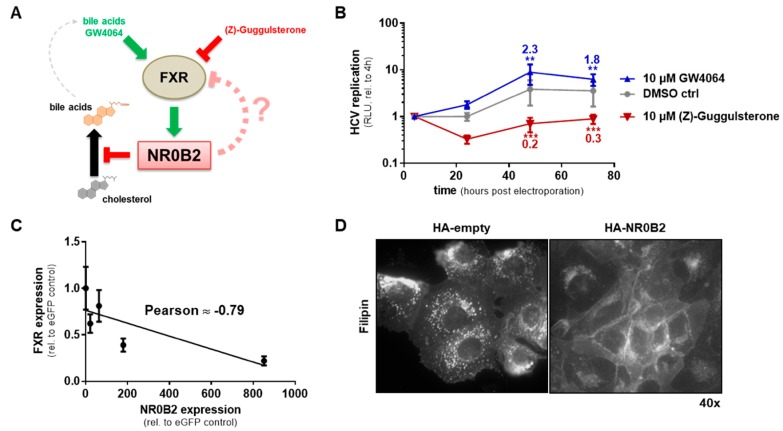
HCV replication strongly depends on an intact cholesterol-bile acid axis, orchestrated by FXR and NR0B2. (**A**) Schematic of the FXR-NR0B2 axis that regulates bile acid synthesis. (**B**) Replication of a subgenomic luciferase reporter replicon (gt1b) after electro-transfection into Huh7-Lunet HA-eGFP cells treated with 10 µM GW4064, (Z)-Guggulsterone, or DMSO as a control in a time-course of 72 h. (**C**) FXR expression levels in Huh7-Lunet cells with different overexpression levels of HA-NR0B2. (**D**) Free cholesterol distribution visualized by Filipin-III staining in HA-NR0B2 overexpressing Huh7-Lunet cells. Luciferase data (**B**) is normalized to the input signal four hours post transfection and shows mean and standard deviation from three independent biological experiments. Significance was calculated using two-tailed, non-paired Student’s *t*-test. ** *p* ≤ 0.005, *** *p* ≤ 0.001.

**Table 1 viruses-12-00036-t001:** Name and sequence of qRT-PCR primers used in this study.

Name	Sequence 5′-3′
THAP7_fwd	ccttagcagccccttttcag
THAP7_rev	ccacctggtagctgtgttca
CRYM_fwd	aaacctcccagcagtgaagt
CRYM_rev	gaccgaagaacagacccgta
LBHD1_fwd	ctcaaaagtcccatctgccg
LBHD1_rev	gacttctagggtcctgtggg
CRAMP1_fwd	gaagaagctgtgcgatccag
CRAMP1_rev	agctccacgatcatcctgag
NR0B2_fwd	ggagcttagccccaaggaat
NR0B2_rev	agggttccaggacttcacaca
FXR_fwd	tgtgaggggtgtaaaggtttct
FXR_rev	gccaacattcccatctctttgc
GAPDH_fwd	tcggagtcaacggatttggt
GAPDH_rev	ttcccgttctcagccttgac

**Table 2 viruses-12-00036-t002:** Name and sequence of PCR primers used for LBHD1 cloning.

Name	Sequence 5′-3′
C11orf48 cDNA 3′-UTR R	ggggcttttgtcttcttttgc
C11orf48 attB F	GGGGACAAGTTTGTACAAAAAAGCAGGCTTCatggcccttgtgccagggagaa
C11orf48 attB R open	GGGGACCACTTTGTACAAGAAAGCTGGGTCgtcctggctggctttgaaggggct

**Table 3 viruses-12-00036-t003:** Name and sequence of PCR primers used for CRAMP1 cloning.

Name	Sequence 5′-3′
CRAMP1 attB F	GGGGACAAGTTTGTACAAAAAAGCAGGCTTCatgacagtgaagttgggcgac
CRAMP1 overlap R	gccttcttccttctcgccaccagaagaccctaagttccg
CRAMP1 overlap F	cggaacttagggtcttctggtggcgagaaggaagaaggc
CRAMP1 attB R open	GGGGACCACTTTGTACAAGAAAGCTGGGTCctgggacaggtcactgacagc

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
