# Peer review of "Gene Expression Profiling of Different Huh7 Variants Reveals Novel Hepatitis C Virus Host Factors"

_viruses, 2019, doi:10.3390/v12010036_

Round 1

Reviewer 1 Report

In this manuscript, Dachert et al. have measured HCV replication efficiency in a range of Huh7 derived cell lines, and performed genome-wide analysis of host gene expression levels. Using this approach, the authors identified host genes that were associated with HCV replication and validated the activity of these genes by over-expressing or silencing them in highly permissive or low permissivity cell lines. They identified two host factors that significantly affected HCV replication.

Overall the work is of interest and seems to have been well done. There were a few points where I would have appreciated a little more explanation, as detailed below. There are also a few instances of non-standard formatting, but they can be easily corrected.

Specific points

The authors referred to previous mathematical modelling suggesting a single limiting factor conferring high or low permissivity to HCV replication, but ultimately reported several putative hits and two that were validated. Could they speculate on the discrepancy between the model and the presented results. Do they suggest that one of the factors is far more important than the other? Could the authors propose any reason why a slight increase in the expression of NR0B2 increases HCV replication in highly permissive Lunet cells, but not in the less permissive Huh7-LP cells? This is counter intuitive as there should be more of a window to increase NR0B2 expression in cells that have only a small amount to begin with.  Similar to point 2, did the authors try the FXR activator/inhibitor experiments in low permissivity cells? If the issue is the difficulty of achieving a small enough over expression of NR0B2, these compounds might provide an alternative means of validation. The difference in permissivity to HCV replication between high and low permissivity cells is ~1000-fold. None of the factors identified seem to account for such a large change. Do the authors think this suggests other as yet unidentified factors, or that several factors play a role and the permissivity is extremely sensitive to small variations in the relative abundance of each of them.  “Mio” on line 38 should probably be written “million” and the use of inverted commas on line 259 is non-standard for English.

Author Response

1) The authors referred to previous mathematical modelling suggesting a single limiting factor conferring high or low permissivity to HCV replication, but ultimately reported several putative hits and two that were validated. Could they speculate on the discrepancy between the model and the presented results. Do they suggest that one of the factors is far more important than the other?

We do think that more than one gene contributes to permissiveness. We realize we have not made this clear enough in the original manuscript: the “limiting host factor” predicted by the mathematical model- as well as by earlier work (Lohmann et al., JVI 2003)- might very well (and very likely) comprise a full, complex signaling or metabolic network. Given the localization within the replication cycle, namely right at the formation of the membraneous replication compartment, a not unlikely possibility would be membrane biogenesis or PIP or cholesterol metabolism, to name just a few. In this case, the product (membranes, PI4P, correctly localized cholesterol, …) would represent the limiting host resource, and many genes might effectively contribute to its production (at a regulatory or the biosynthetic level). We have now extended our discussion on this, both, in the introduction (lines 85ff), as well as in the discussion section (line 945ff).

2) Could the authors propose any reason why a slight increase in the expression of NR0B2 increases HCV replication in highly permissive Lunet cells, but not in the less permissive Huh7-LP cells? This is counter intuitive as there should be more of a window to increase NR0B2 expression in cells that have only a small amount to begin with.

The reviewer is absolutely correct– we would have expected this as well. Unfortunately, the slightest overexpression we could achieve in Huh7-LP cells was 44x, while Lunet only contain 2.3x more NR0B2 (see modified figure 6B); this suggests overexpression in Huh7-LP might already be beyond the beneficial levels. Moreover, additional limitations likely exist in Huh7-LP that prevent NR0B2 from fulfilling its role, see our discussion to point 1) above and 3) below.

3) Similar to point 2, did the authors try the FXR activator/inhibitor experiments in low permissivity cells? If the issue is the difficulty of achieving a small enough over expression of NR0B2, these compounds might provide an alternative means of validation.

This is a great suggestion and we have now performed these experiments. Regrettably, GW4064 treatment did not increase HCV replication in Huh7-LP cells in the infection setting. This surprising result prompted us to check expression levels of FXR in Huh7-LP cells, and much to our astonishment, we found substantial differences between the two Huh7-LPs (p13, p28) and Huh7-Lunet (aprx. 4-fold, see transcriptomic data). This might explain why, both, NR0B2 overexpression and FXR agonist treatment in these cells (low FXR, low NR0B2) might have a different outcome from Lunet cells (4x higher FXR, 2x higher NR0B2). It corroborates our notion that it really is the balance of the FXR/NR0B2 axis that determines HCV permissiveness. We decided not to show this negative data but have included it in our discussion on this matter (lines 1128ff).

4) The difference in permissivity to HCV replication between high and low permissivity cells is ~1000-fold. None of the factors identified seem to account for such a large change. Do the authors think this suggests other as yet unidentified factors, or that several factors play a role and the permissivity is extremely sensitive to small variations in the relative abundance of each of them.

This is an important question, and we try to be clearer about this issue in the manuscript now– please see our reply to point 1 above! Taken together, we do not think it is a contradiction to the mathematical model that more than one gene contributes to the permissiveness phenotype. Therefore, it is feasible or even likely that- as the reviewer reasoned- small variations in the abundance of several genes together are required to provide the optimal environment for HCV (e.g. the balance of FXR and NR0B2 expression, see point 3). However, it is beyond the scope of this (already rather extensive) study to test combinatorial effects of different expression levels of different genes.

5) “Mio” on line 38 should probably be written “million” and the use of inverted commas on line 259 is non-standard for English.

Thank you for pointing this out– we corrected all occurrences.

Reviewer 2 Report

HCV is major human pathogen and remains a great public health concern across the world. There exists highly effective antiviral drugs but no effective vaccine has been developed. Further, HCV is the type species of a genus Hepacivirus that infect a wide range of animals, and is a relatively close relative to viruses in the Flaviviridae family like Zika or dengue viruses who are also important pathogens. Understanding factors affecting viral replication - although important scientifically in their own right - may aid production/development of HCV vaccines or novel drugs targeting related viruses.

Dachert et al., report a series of experiments following on from their previous PLoS Pathogens paper (Binder et a., 2013) suggesting that a previously undescribed single factor played a major role in differential regulation of HCV replication in vitro – this factor was likely limiting in poorly permissive cells and abundant in highly permissive cells. IN this well-written manuscript they detail their large body of work on identifying novel host factors regulating HCV replication in tissue culture. The approach used combines comparative gene expression analysis of different clones of a single cell line Huh7 with variable permissivity to HCV replication and downstream functional validation (loss and gain of function) of these hits. Here they provide evidence of a few new host genes that appear to regulate HCV replication in Huh7 cells.

While for one gene NR0B2, Dachert et al., carry out significant further work establishing a role for this gene during HCV replication, there is high degree of loss of their hits (most hits that looked promising initially, could not be proven to play a major role in functional experiments). This is perhaps not surprising given their work is based on comparative gene expression correlation to HCV replication and for many genes there is not a good correlation between mRNA expression and functional protein abundance. For NR0B2, it appears that it has a measurable role during HCV replication and that its function may lie in the known association between HCV replication and lipid metabolism in the host cell. However, it alone likely does not explain the initial observation of major differences in replication capacity of Huh7 lines for HCV.

To make the case for this approach and these genes stronger a number of things should be done:

Fig1A – did this data from the previous publication? If so, this should be stated clearly in the figure legend. If not, more description is required. Further, is there error bars for this data? What is the rationale for choosing a cut off between high or low permissivity? Fig1C – At least for the genes followed up on, it would be worthwhile to show the non-relative levels of transcript in each cell line from Fig1A. How much basic characterisation of these different cell lines has been done? Do they look or grow differently from each other? Do you think this would affect replication of HCV? Typically, in transfection experiments for HCV cells are seeded at low density and allowed to grow over 2-3days. Luciferase-based HCV experiments do not typically have a control for general gene expression or protein synthesis over time and so differences in cell growth may influence accumulation of reporter protein and obscure true results. Can the authors comment on whether this cold influence interpretation of early or later results? As the raw data is not provided in this manuscript it is difficult to assess this aspect. It would be worthwhile to make this easily available for interested parties to see how their gene of interest correlates with HCV replication. Fig2 A – why is this in Z score format? Please put it in RLU rel to GFP like what is done for Fig2 B. Would it be useful to include the complete data set for Fig2A? Although I see this is in Z score format in the supplement.  Fig 3 – why do you think N-term HA and C-term flag tagging has disparate affects on HCV replication? Do you believe that they do differentially affect protein function or do you think that the small fold-changes represent noise in the assay and so likely these genes have no effect? This would be worthwhile to know for anyone working on CRYM/LBHD1 or CRAMP1 in the future. Additionally, in some of the panels it looks like it is the negative control cells that change the most between HA or FLAG (from what I can see in A and D). Is this a correct interpretation of this data? The authors should be commended in showing the seemingly negative data on hits that did not appear to play a role during HCV replication. The negative data from certain genes makes the positive data of THAP7 and NR0B2 more convincing. Throughout – what is the rationale for tagging proteins with epitope tags but never showing blots to the epitope tags? Surely doing this would allow for better comparison of expression levels of proteins between genes? Fig 5: is the house keeping protein alpha or beta actin or both? Why in B is it just above 37kda and in F it is below the 37 kda mark. Please comment/fix. The Nr0B2 work is certainly the most interesting aspect of this manuscript as it appears to play an interesting role during HCV replication in Huh7 cells. I agree with the model proposed in D – especially if you consider that siRNA mediated knockdown in highly permissive Lunet line seemingly reduced replication while over-expression also reduced replication Are the authors able to show NR0B2 mRNA levels in lowly expressing cell lines compared to the overexpression panel set> 7C – is it possible to do formal correlation analysis on your data in this case? Does over expression of NR0B2 affect cell growth or morphology? Is it affecting HCV specifically or is this a more general phenomenon? Is there a virus that does not utilise lipid metabolism to the same degree as HCV that could be used as a control. Although I understand virus and host cell growth in tightly linked in many instances. While there is ample discussion surrounding how the genes might effect or not effect HCV replication, there is little general discussion of how well this experimental analysis fits with the proposed model put forward in Binder et al 2013 paper. On the whole it looks like while the authors found a novel host factor in Nr0B2 and perhaps THAP7 it is not likely the host factor they were searching for. Does this mean that the proposed model needs to change or that searching for the factor by comparative gene expression analysis is not the ideal means? Indeed having two genes controlling replication even goes against the single gene hypothesis.  As the authors are experts on viral/innate immune interactions they may have thought of this but I know that this would not fit with the computer model but perhaps the authors could search for restriction factors in their data (i/e those that negatively correlate with HCV replication). Differences in RNA sensing or even HCV inhibitors could influence these experiments and form the basis of fascinating future studies. There is precedent for this in the loss of Rigi signalling capacity in Huh7 cell lines. The resource of Huh7 lines with gene expression correlating to HCV replication will be an interesting venture for future investigations. 

Author Response

HCV is major human pathogen and remains a great public health concern across the world. There exists highly effective antiviral drugs but no effective vaccine has been developed. Further, HCV is the type species of a genus Hepacivirus that infect a wide range of animals, and is a relatively close relative to viruses in the Flaviviridae family like Zika or dengue viruses who are also important pathogens. Understanding factors affecting viral replication - although important scientifically in their own right - may aid production/development of HCV vaccines or novel drugs targeting related viruses.

Dachert et al., report a series of experiments following on from their previous PLoS Pathogens paper (Binder et a., 2013) suggesting that a previously undescribed single factor played a major role in differential regulation of HCV replication in vitro – this factor was likely limiting in poorly permissive cells and abundant in highly permissive cells. IN this well-written manuscript they detail their large body of work on identifying novel host factors regulating HCV replication in tissue culture. The approach used combines comparative gene expression analysis of different clones of a single cell line Huh7 with variable permissivity to HCV replication and downstream functional validation (loss and gain of function) of these hits. Here they provide evidence of a few new host genes that appear to regulate HCV replication in Huh7 cells.

While for one gene NR0B2, Dachert et al., carry out significant further work establishing a role for this gene during HCV replication, there is high degree of loss of their hits (most hits that looked promising initially, could not be proven to play a major role in functional experiments). This is perhaps not surprising given their work is based on comparative gene expression correlation to HCV replication and for many genes there is not a good correlation between mRNA expression and functional protein abundance. For NR0B2, it appears that it has a measurable role during HCV replication and that its function may lie in the known association between HCV replication and lipid metabolism in the host cell. However, it alone likely does not explain the initial observation of major differences in replication capacity of Huh7 lines for HCV.

As a general remark, we want to point out that we now tried to clarify in the introduction and discussion sections that we do not necessarily expect “host factor” to refer to one single gene explaining the full 1000-fold difference in permissiveness– please also see our response to reviewer 1 and our additional discussion in the manuscript lines 85ff and 945ff.

To make the case for this approach and these genes stronger a number of things should be done:

Fig1A – did this data from the previous publication? If so, this should be stated clearly in the figure legend. If not, more description is required. Further, is there error bars for this data? What is the rationale for choosing a cut off between high or low permissivity?

Yes, panel A is from the previous publication with slight modifications (inverted order of samples) and this is stated in the figure legend. To make it even clearer, we moved the statement from the last sentence of the legend right into the description of panel A where we now also give more detail on the measurement. Note that this data stems from exactly one biological replicate, as we extracted the RNA for microarray analyses from exactly these cells. Technical replicates (triplicate wells) showed negligible variance why error bars would hardly be visible and were omitted. The general observation that different passages of Huh7 or subclones show this degree of difference in HCV replication has been established thoroughly before (e.g. Blight et al. 2002, Lohmann et al 2003, Binder et al. 2007). We are unsure if we understood the reviewer’s question about a “cut off between high and low permissivity”– we used this binary distinction only for one of our correlation criteria (also shown on the X-axis in panel C) in which we make sure that “highly permissive” cells have a higher expression than “lowly permissive” cells, i.e. expression positively correlates with permissiveness. This is mentioned in the methods section. Furthermore, we use the term “Huh7-LP” and “Huh7-HP” in the manuscript to refer to “low passage” (LP) and “high passage” (HP) number, NOT “low permissiveness” and “high permissiveness”. We now introduce both terms in the “Cell lines” section of the materials and methods.

Fig1C – At least for the genes followed up on, it would be worthwhile to show the non-relative levels of transcript in each cell line from Fig1A.

This is a valuable suggestion and we have now plotted the non-relative signal strengths from the microarray analysis for the followed-up genes in the new panel 1D.

How much basic characterisation of these different cell lines has been done? Do they look or grow differently from each other? Do you think this would affect replication of HCV? Typically, in transfection experiments for HCV cells are seeded at low density and allowed to grow over 2-3days. Luciferase-based HCV experiments do not typically have a control for general gene expression or protein synthesis over time and so differences in cell growth may influence accumulation of reporter protein and obscure true results. Can the authors comment on whether this cold influence interpretation of early or later results? As the raw data is not provided in this manuscript it is difficult to assess this aspect. It would be worthwhile to make this easily available for interested parties to see how their gene of interest correlates with HCV replication.

Lohmann et al., JVI 2003, have characterized various aspects of differently permissive Huh7 passages and found no obvious relation to, e.g., their translational capacity. They (and we: Binder et al., PLoS Path 2013) have also performed HCV RNA quantification by Northern blotting over time in which HCV levels were normalized to a cellular transcript to control for differential cell growth. We furthermore now provide microscopy images for the reader to appreciate the slight differences in cell morphologies of three different Huh7 variants (Huh7-LP, -HP and -Lunet), as well as comparative growth curves for the same set of cells (new Figure 1B and S2). We now also discuss these aspects briefly in the results section in lines 308ff.

Fig2 A – why is this in Z score format? Please put it in RLU rel to GFP like what is done for Fig2 B. Would it be useful to include the complete data set for Fig2A? Although I see this is in Z score format in the supplement. 

We fully agree with the reviewer that this may cause confusion; we now show RLU relative to GFP also in panel 1A. The reason for z-scores was the somewhat higher robustness of this measure, particularly when combining several (here: 3) independent repetitions of a high number of conditions (here: >100 siRNAs). However, as variance was limited in this case, we now also changed the supplementary figure to “relative RLU” to keep everything consistent (Figure S10).

Fig 3 – why do you think N-term HA and C-term flag tagging has disparate affects on HCV replication? Do you believe that they do differentially affect protein function or do you think that the small fold-changes represent noise in the assay and so likely these genes have no effect? This would be worthwhile to know for anyone working on CRYM/LBHD1 or CRAMP1 in the future.

Disparate effects of the differently tagged constructs can have various reasons, none of which could be easily tested. We have included a brief discussion on the topic right in the results (lines 423ff) now. However, we deem it very unlikely that the phenomenon was sheer noise, as the data presented in this figure come from at least three independent repetitions of the experiment, always with the same (or comparable) outcomes. Furthermore, we have now performed further experiments using infection with the JcR2a luciferase reporter virus, whose replication was also rescued by overexpression of the proteins in Huh7-LP cells (Figure 3C,F,I), corroborating the positive effect we have observed for the subgenomic replicon. We have therefore now rephrased this results section and do not dismiss CRYM and LBHD1 as unreliable, but in fact, as two factors possibly contributing to host cell permissiveness (paragraph starting at line 391 has been largely rewritten).

Additionally, in some of the panels it looks like it is the negative control cells that change the most between HA or FLAG (from what I can see in A and D). Is this a correct interpretation of this data?

This observation is correct. Based on her/his previous comments we assume the reviewer is familiar with HCV replication assays and can confirm that absolute numbers can hardly be reproduced in independent experiments, but relative trends are very robust. For this reason, we have always run passage-matched overexpression and corresponding (same tag) vector control cells in parallel, using the same batch of in vitro transcribed RNA per experiment. While absolute RLU naturally varied between repetitions (see error bars), we very much trust the relative difference between overexpression and control cells.

The authors should be commended in showing the seemingly negative data on hits that did not appear to play a role during HCV replication. The negative data from certain genes makes the positive data of THAP7 and NR0B2 more convincing. Throughout – what is the rationale for tagging proteins with epitope tags but never showing blots to the epitope tags? Surely doing this would allow for better comparison of expression levels of proteins between genes?

We thank the reviewer for her/his appreciation of our showing of negative results as well– we share her/his opinion that negative data should also be reported and if only to highlight the significance of the positive ones. We furthermore now include the anti-FLAG and anti-HA western blots for all constructs in Figure S4D and S4E). We previously simply focused on the two hottest candidates (THAP7 and NR0B2) and for these we had at some point decided to buy specific antibodies. But we agree, we have the blots, we show the blots.

Fig 5: is the house keeping protein alpha or beta actin or both?

It is beta-actin, the alpha-sign was used as the common abbreviation for “anti-“, but we agree this looks confusing in this instance. We now use brackets to make this clearer; the figure has been moved to the supplements (Figure S6D and E).

Why in B is it just above 37kda and in F it is below the 37 kda mark. Please comment/fix.

This is again a good point– we were originally not sure how to handle this case: the gel /blot was not straight and we simply indicated the height of the marker band at the left-hand side of the gel. We have now interpolated the marker height and hope this is an appropriate handling of the issue. For the reviewers’ and the editor’s reference, this is the original image of the beta-actin blot (regular photo overlayed with luminescence signal): [please see provided PDF of this point-by-point reply]

The Nr0B2 work is certainly the most interesting aspect of this manuscript as it appears to play an interesting role during HCV replication in Huh7 cells. I agree with the model proposed in D – especially if you consider that siRNA mediated knockdown in highly permissive Lunet line seemingly reduced replication while over-expression also reduced replication Are the authors able to show NR0B2 mRNA levels in lowly expressing cell lines compared to the overexpression panel

Yes, we now show the mRNA level in lowly permissive cells next to Huh7-Lunet and the overexpression lines. This information was previously “hidden” in the old Figure 5E (now S6C/F), but it is definitely a very good suggestion to put it into Figure 6B as well!

set> 7C – is it possible to do formal correlation analysis on your data in this case?

We have done linear correlation analysis on this dataset and yielded a Pearson coefficient of -0.79. We have included this now in line 887f and in the figure panel (now Figure 8C).

Does over expression of NR0B2 affect cell growth or morphology?

We have performed cell growth assays and took microscopic images for the reviewer and the reader to appreciate the differences (Figure S7). At very high expression levels, morphology changes more obviously but cell growth is not substantially affected.

Is it affecting HCV specifically or is this a more general phenomenon? Is there a virus that does not utilise lipid metabolism to the same degree as HCV that could be used as a control. Although I understand virus and host cell growth in tightly linked in many instances.

We agree that this is an exciting question and we have now performed replication experiments using Dengue virus (a closer relative of HCV) and Rift valley fever virus (a non-related negative-strand / ambisense RNA virus). Both viruses are largely unaffected by NR0B2 overexpression, with the slight effects on Dengue virus resembling the tendency seen for HCV (see new Figure 7). This very much corroborates our notion of an HCV-specific effect of NR0B2, although we cannot rule out that other highly lipid-dependent viruses might rely on it as well. We think our manuscript has benefitted greatly from this experiment and we thank the reviewer for suggesting it!

While there is ample discussion surrounding how the genes might effect or not effect HCV replication, there is little general discussion of how well this experimental analysis fits with the proposed model put forward in Binder et al 2013 paper. On the whole it looks like while the authors found a novel host factor in Nr0B2 and perhaps THAP7 it is not likely the host factor they were searching for. Does this mean that the proposed model needs to change or that searching for the factor by comparative gene expression analysis is not the ideal means? Indeed having two genes controlling replication even goes against the single gene hypothesis. 

The reviewer is absolutely right in that we have 1) not well enough explained the meaning of the term “one single host factor” and 2) not appropriately discussed the results on the background of the model prediction. We tried to improve on both now: ad 1) we have now added more explanation on this in the introduction (lines 85ff), please also see our comments to reviewer 1. Ad 2), we have now included a more extensive discussion on this topic (lines 945ff). In brief, we do think it is very well possible that the “single factor” limiting replication in lowly permissive cells can also be a complex pathway / process, e.g. membrane biosynthesis or cholesterol metabolism. Our reasoning was that availability of a limiting resource (e.g. membrane or cholesterol) may or should have some transcriptional basis. It would have been nice to find a single gene whose product is of too limited abundance for HCV to replicate efficiently, but as it stands, chances are several / many factors contribute to the availability of this limiting resource, and, possibly, we have identified a few of the contributing pathways.

As the authors are experts on viral/innate immune interactions they may have thought of this but I know that this would not fit with the computer model but perhaps the authors could search for restriction factors in their data (i/e those that negatively correlate with HCV replication). Differences in RNA sensing or even HCV inhibitors could influence these experiments and form the basis of fascinating future studies. There is precedent for this in the loss of Rigi signalling capacity in Huh7 cell lines. The resource of Huh7 lines with gene expression correlating to HCV replication will be an interesting venture for future investigations. 

This is indeed a very interesting aspect the reviewer brings up here. In fact, many years ago we have looked for inhibitory factors in lowly permissive cells as well, but could not easily find any. We further put some trust in the findings of Lohmann et al. (JVI 2003), who concluded from titration experiments that a positive rather than a negative factor would govern permissiveness, although this was only hints, not proof. Lastly, for the mentioned precedence of RIG-I, we have done extensive research and could not show any impact on cellular permissiveness at all (Binder et al., Hepatology 2007), although we are aware of somewhat opposing findings from Michael Gale’s team. We, hence, now focused on the hypothesis of a limiting positive factor, which was also supported by our modelling work (PLoS Pathogens 2013), but we offer the full transcriptomic data (GEO accession given in the manuscript) for the community to go ahead and analyze it in any other direction. We included some discussion on the possibility of negative regulators now in lines 938ff (and touch upon it in lines 85ff in the introduction).

Reviewer 3 Report

In the manuscript entitled “Gene Expression Profiling of  Different Huh7 Variants Reveals Novel Hepatitis C Virus Host Factors”, the authors identified some host cellular factors contributing to HCV permissiveness by confronting full-genomic expression profiles across a panel of eight differently permissive Huh7-based cell lines. In particular, they found two transcriptional regulators, THAP7 and NR0B2, whose expression levels correlate with HCV replication efficiency. This study provides relevant contribution to a better understanding of the molecular host factors responsible for HCV host cell tropism, that is important  to develop new cell culture systems for studying  HCV pathogenesis and treatment. Overall the study is of interest for the readership of  Viruses, the research design is appropriate,  the results clearly presented and the manuscript nicely crafted. However, in my opinion, some points should be addressed to make the manuscript more solid.

 Main concern:

- Authors should better justify their statement regarding the poor effect of CRYM expression levels on HCV replication. HA- CRYM over-expression in Huh7-LP cells induced a 2,32-fold increase in HCV replication, that is similar to the 2,5 fold increase in HCV replication caused by LBHD1 over-expression. In my opinion, also HA-CRYM might contribute to cellular permissiveness for HCV and both factors should be analyzed more in depth. For example , authors should test the effects of CRYM and LBHD1 over-expression on replication efficiency of other genotypes, using for example gt2a replicon.

- Could THAP7 and NR0B2 overexpression render HUH7-LP cells  permissive to JcR2a HCV infection? Could  you add data about this point?

Author Response

In the manuscript entitled “Gene Expression Profiling of  Different Huh7 Variants Reveals Novel Hepatitis C Virus Host Factors”, the authors identified some host cellular factors contributing to HCV permissiveness by confronting full-genomic expression profiles across a panel of eight differently permissive Huh7-based cell lines. In particular, they found two transcriptional regulators, THAP7 and NR0B2, whose expression levels correlate with HCV replication efficiency. This study provides relevant contribution to a better understanding of the molecular host factors responsible for HCV host cell tropism, that is important  to develop new cell culture systems for studying  HCV pathogenesis and treatment. Overall the study is of interest for the readership of  Viruses, the research design is appropriate,  the results clearly presented and the manuscript nicely crafted. However, in my opinion, some points should be addressed to make the manuscript more solid.

Main concern:

- Authors should better justify their statement regarding the poor effect of CRYM expression levels on HCV replication. HA- CRYM over-expression in Huh7-LP cells induced a 2,32-fold increase in HCV replication, that is similar to the 2,5 fold increase in HCV replication caused by LBHD1 over-expression. In my opinion, also HA-CRYM might contribute to cellular permissiveness for HCV and both factors should be analyzed more in depth. For example , authors should test the effects of CRYM and LBHD1 over-expression on replication efficiency of other genotypes, using for example gt2a replicon.

We agree with the reviewer in that we probably were too negative about the effects of CRYM and LBHD1. While not strong, the enhancing effect on HCV replication was clearly reproducible across biological experiments and the fact that C-terminal FLAG-tagging lost this effect could have many (also technical) reasons. We have furthermore gladly followed the reviewer’s suggestion to validate the effect of these two genes by another genotype. We went for infection with a gt2a reporter virus (JcR2a) and, in fact, could confirm the positive effect of the genes’ expression in lowly permissive cells. While JcR2a hardly replicates in Huh7-LP cells, expression of the genes rescued its replication significantly (see new figure 3C,F,I). We therefore have rephrased the corresponding results section (paragraph starting at line 391 was largely rewritten) and more clearly state now that all three, CRYM, LBHD1 and THAP7, have the capacity to increase host cell permissiveness towards HCV. We want to point out that for THAP7 the rescue in the infection setting was more than one log, which we find quite impressive. We thank the reviewer for this suggestion not to “undersell” these factors!

- Could THAP7 and NR0B2 overexpression render HUH7-LP cells  permissive to JcR2a HCV infection? Could  you add data about this point?

As stated above, we have now added experiments using JcR2a infection for all candidate overexpressions. For NR0B2 overexpressing cells, analogously to the experiments using the subgenomic gt1b replicon, we found replication of JcR2A to suffer substantially when expressed in Huh7-Lunet (see new figure 7). As for Huh7-LP cells, NR0B2 overexpression if anything inhibited JcR2A replication similar to the replicon setting (see new Figure 5B). We think it is possible that we could not achieve low enough overexpression for NR0B2 in LP cells and/or the lack of FXR expression in these cells prevents NR0B2 from functioning; please also see our comment to point 3) of reviewer 1 on this topic.

Round 2

Reviewer 3 Report

In the manuscript entitled "Gene Expression Profiling of Dfferent Huh7 Variants reveals Novel Hepatitis C Virus Host Factors" the authors identified four host factors playing a major role in regulating  HCV replication in vitro. They carried out a large body of work to demonstrate the role of these factor in HCV permissiveness of Huh7 based cells. I believe that this study provides interesting results that might facilitate HCV culture  in vitro that is essential for  developing new HCV drugs.

Furthermore, the authors  replied in an exhaustive manner to  all of my observations. 

To my point of view the manuscript  can be published in "Viruses".